# Dynamic computational phenotyping of human cognition

Roey Schurr ®[1,7] ✉, Daniel Reznik ®[2,7] ✉, Hanna Hillman ®[3], Rahul Bhui ®[4,5] & Samuel J. Gershman[1,6]

Computational phenotyping has emerged as a powerful tool for characterizing individual variability across a variety of cognitive domains. An individual's computational phenotype is defined as a set of mechanistically interpretable parameters obtained from fitting computational models to behavioural data. However, the interpretation of these parameters hinges critically on their psychometric properties, which are rarely studied. To identify the sources governing the temporal variability of the computational phenotype, we carried out a 12-week longitudinal study using a battery of seven tasks that measure aspects of human learning, memory, perception and decision making. To examine the influence of state effects, each week, participants provided reports tracking their mood, habits and daily activities. We developed a dynamic computational phenotyping framework, which allowed us to tease apart the time-varying effects of practice and internal states such as affective valence and arousal. Our results show that many phenotype dimensions covary with practice and affective factors, indicating that what appears to be unreliability may reflect previously unmeasured structure. These results support a fundamentally dynamic understanding of cognitive variability within an individual.

Untangling sources of individual variability remains a central challenge in cognitive science. This endeavour has been revolutionized by the use of computational models[1], which provide precise algorithmic accounts of cognitive processes in terms of parsimonious sets of parameters, collectively termed the 'computational phenotype'[2]. Importantly, these computational parameters can be intuitively interpreted as cognitively meaningful entities, such as learning rate or risk attitude. The interpretability of the computational phenotype has made it an appealing tool for studying complex phenomena as far-reaching as brain function[3–6], psychiatric illness[7,8], developmental processes[9–11] and cross-species variation[12,13]. For example, research in the field of computational psychiatry demonstrates that computational modelling can be particularly insightful for teasing apart

different behavioural aspects of mental illness. While the link between anxiety and disrupted decision-making is well established[14], characterizing the specific behavioural disruption was accomplished in a study that estimated the computational phenotype in patients diagnosed with pathological anxiety and healthy controls. The study showed that anxiety is specifically associated with enhanced risk aversion (indicating less risk-taking) but not loss aversion[15] (see also ref. 16). Another example of the merits of computational phenotyping comes from developmental science. Previous behavioural studies have shown that children tend to explore more than adults[17]. While this could be explained by generally noisier behaviour in children, a computational phenotyping study helped to elucidate this phenomenon, indicating that in fact children rely more on directed, but not random exploration,

[1]Department of Psychology, Center for Brain Sciences, Harvard University, Cambridge, MA, USA. [2]Department of Psychology, Max Planck Institute for Human Cognitive and Brain Sciences, Leipzig, Germany. [3]Department of Psychology, Yale University, New Haven, CT, USA. [4]Sloan School of Management, Massachusetts Institute of Technology, Cambridge, MA, USA. [5]Institute for Data, Systems, and Society, Massachusetts Institute of Technology, Cambridge, MA, USA. [6]Center for Brains, Minds, and Machines, Massachusetts Institute of Technology, Cambridge, MA, USA. [7]These authors contributed equally: Roey Schurr, Daniel Reznik. ✉e-mail: roey.schurr@mail.huji.ac.il; reznik@cbs.mpg.de

thereby reducing uncertainty about the environment by choosing high-uncertainty options[18].

Despite the widespread use of computational phenotypes, their interpretation hinges critically on their psychometric properties[19], which remain poorly understood[20]. This issue is even more prominent in longitudinal studies that address changes within individuals over time[21,22]. However, test-retest reliabilities of the computational phenotypes remain largely unknown since computational models are rarely fit within the same subjects over more than one timepoint. Only a few studies explicitly address the reliability of computational phenotypes (see review in ref. [20]) and rarely in more than two sessions (but see ref. [23]). Such studies have found mixed results, with most phenotype parameters showing poor test-retest reliability and a few showing moderate to high test-retest reliability. Furthermore, a large-scale study focusing on the domain of self-control showed significantly lower reliability for task-based measurements, including the computational phenotype, compared with classic self-reported measurements[24].

Low test-retest reliability of the computational phenotypes could reflect measurement noise, non-stationarity of the underlying construct, or both. If the underlying construct is non-stationary, its temporal trajectory could be relatively unpredictable (for example, a random walk) or relatively predictable (for example, directional drift induced by practice). Deciphering these sources of variability necessitates a robust, high-powered longitudinal dataset—a task that we undertake in this study. Our investigation seeks to better discern the 'noise' and the 'signal' in computational phenotypes by modelling multiple potential sources of temporal variability.

Over a continuous 3-month period, we engaged 90 human participants in a weekly battery of 7 online computer-based tasks: Go/No-go, Change detection, Random dot motion, Lottery ticket, Intertemporal choice, Two-armed bandit and Numerosity comparison. These tasks were chosen since they cover various aspects of cognition such as learning, memory, perception and decision making. Using these tasks, we estimated the computational phenotype of each participant on a weekly basis. In addition, the inclusion of a survey tracking individuals' mood and daily activities enabled us to estimate day-specific state effects on the computational phenotype. This unique dataset, which we make publicly available, allows us to illuminate the processes governing the temporal variability of cognition.

Our results provide evidence for a fundamentally dynamic view of the computational phenotype within an individual, and indicate that both practice and affective effects contribute to its temporal variability.

## Results
### Longitudinal data for dynamical computational phenotypes
We collected data from 90 participants who performed 7 online cognitive tasks on a weekly basis for 12 consecutive weeks (Fig. 1). The tasks we used were Go/No-go, Change detection, Intertemporal choice, Lottery ticket, Numerosity comparison, Two-armed bandit and Random dot motion. These tasks were selected for two main reasons. First, they are commonly used in cognitive and neurocognitive research, as well as in phenotyping individual, clinical and age-related variation[25–31]. Second, these tasks have well-established and validated computational models.

First, we calculated reaction time and accuracy (where applicable) for each task averaged (±s.d.) across weeks and participants. As can be seen in Supplementary Table 1, on average, participants' performance was adequate and in line with previous studies using similar tasks. For more detailed analysis of the behavioural data, see Supplementary Fig. 6.

Next, for each participant and for each task, we fit the free parameters with previously validated computational models using a hierarchical Bayesian framework, which formalized various assumptions about within- and between-participant variability[32,33]. By accounting for the structure of the data, this hierarchical framework has been shown to improve parameter stability and provide a more accurate estimate of parameter values at the participant level[33,34]. In particular, we fit two

statistical models to the data: an 'independent' model and a 'dynamic' model (as well as a 'reduced' independent model; see Supplementary Information). The independent model allowed us to quantify parameter stability without building it into our modelling assumptions: the parameters for each participant were assumed to be drawn independently each week from a participant-specific distribution. The dynamic model, which we describe in detail below, formalizes a more structured set of assumptions about how the computational phenotype evolves over time, thus allowing us to make insightful inferences about sources of its temporal variability. Model fitting yielded week-specific estimates for the 19 parameters comprising the computational phenotype for each participant. First, we examined widely used diagnostic measures, such as R-hat and the number of divergent transitions, that serve to assess the convergence of the Markov chain Monte Carlo sampling procedure for parameter estimation (see Supplementary Information). Second, we verified that the parameters were identifiable. Third, we verified that all computational models yielded excellent posterior predictive checks (Supplementary Fig. 8).

We then asked whether the parameter estimates were stable over time within an individual. This was quantified using intraclass correlations (ICC), a widely used measure of test-retest reliability. Figure 2 shows the ICC values for the computational phenotype estimated using the independent model and the reduced model (see Supplementary Fig. 1 for an alternative calculation of the ICC and Supplementary Fig. 2 for a visualization of the variance components used in calculating the ICC). ICC values covered a wide range of 0.49–0.99, with half of the parameters showing poor-to-moderate stability and half moderate-to-excellent stability (according to ref. [35]). In agreement with previous work[34], models with fewer parameters tended to be more stable and parameters derived from the same task tended to have similar values. Go/No-go is a notable counterexample, including both the most stable and least stable parameters across tasks.

While these ICC values are imperfect, indicating variability of the measured computational phenotype over time, they are relatively high compared with those often reported in the literature[20]. We suspect that these relatively high values can be attributed to fitting our data using hierarchical Bayesian modelling, which adequately captures the hierarchical structure of the data. Indeed, previous work showed that the fitting procedure has notable effects on parameter stability, whereby hierarchical models that pool information across participants promote parameter stability[20,32–34]. To test this hypothesis, we repeated the ICC analysis, this time fitting the behavioural data using a reduced hierarchical model. In this reduced model, sessions were not nested within participants; instead, all sessions across all participants were considered independent, such that model parameters were drawn from a single population-level distribution. As expected, this procedure resulted in lower ICC values across all phenotype parameters (Fig. 2, red dots; for further details on this analysis, see Supplementary Information).

We used simulated data to calculate an ICC upper bound for each parameter on the basis of a ground-truth phenotype that was fixed across time (see Methods). This analysis yielded near-perfect stability for all parameters across tasks (Fig. 2, red vertical lines). While such near-perfect stability may seem surprising, it is the result of using the independent hierarchical model in the process of parameter estimation (see Fig. 2 and Supplementary Fig. 4 for lower ICC values obtained from the reduced model, that is, without accounting for the full hierarchical structure of the data). This result indicates that the lower stability values observed in the real data are not the result of low interparticipant variability or of inadequate task design (for example, a low number of trials), but rather that there is true longitudinal variability in the computational phenotype within participants.

Finally, for each task, we also calculated ICC values for the behavioural measures of accuracy and mean reaction time (Supplementary Fig. 5). These values were mostly in the moderate range (0.5–0.75). Reaction times were consistently more stable than accuracy values.

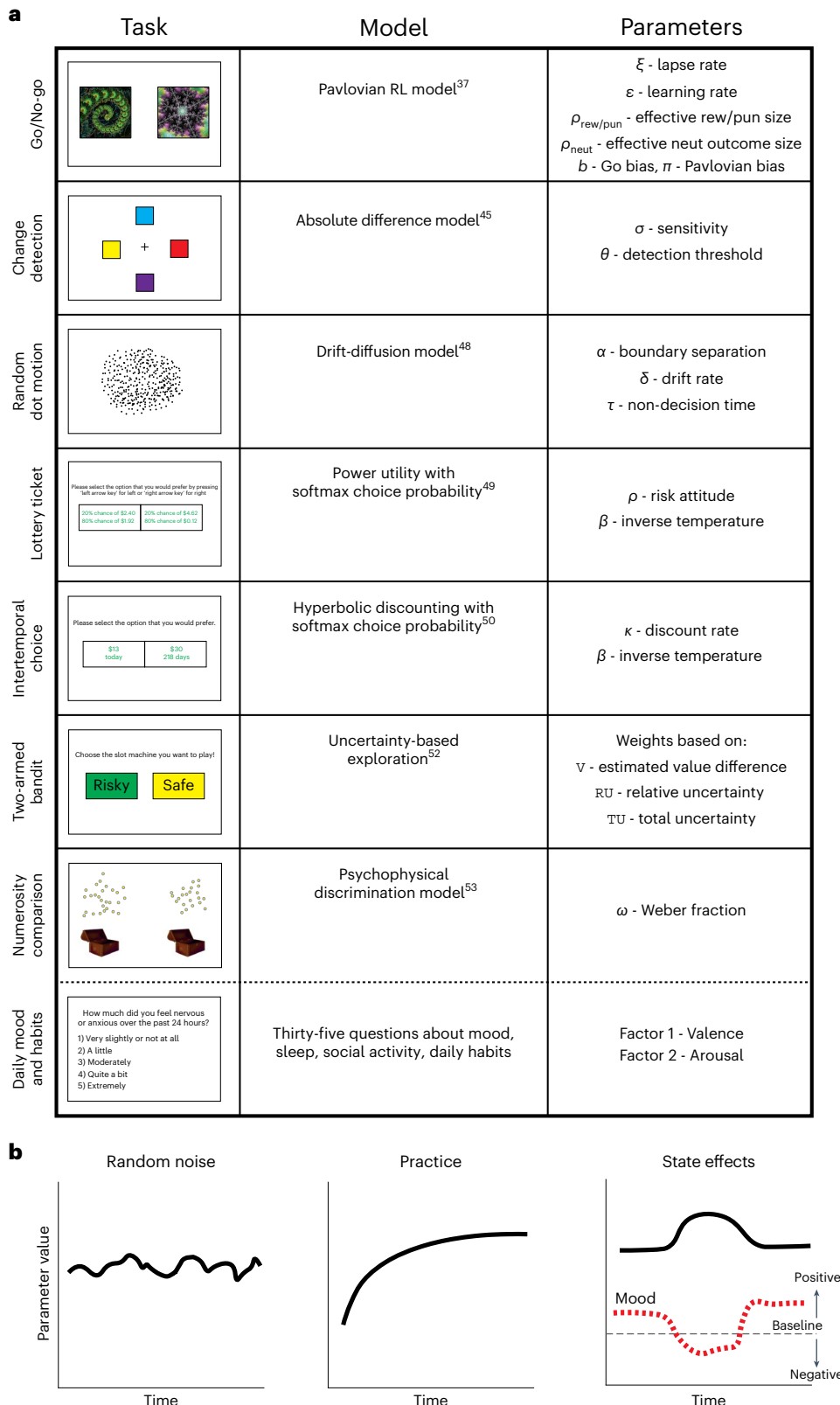

**Fig. 1 | Experimental tasks and models. a**, Participants performed 7 cognitive tasks and a survey on a weekly basis for 12 consecutive weeks. The left column shows an example screen from each task. The middle column shows the computational model used to fit participants' behavioural data from each task. The right column shows the free parameters that we fit to individual participants. The collection of these parameters constitutes the computational phenotype. **b**, Three potential sources of temporal variability: random noise, practice and state effects reflecting changes in mood and daily activities.

For each task, at least some of the computational phenotype parameters were characterized by higher ICC values compared with these summary statistics.

In what follows, we demonstrate our approach for investigating the dynamics of the computational phenotype by giving a detailed account of a single task. We chose the Go/No-go task (Figs. 3 and 4) because this

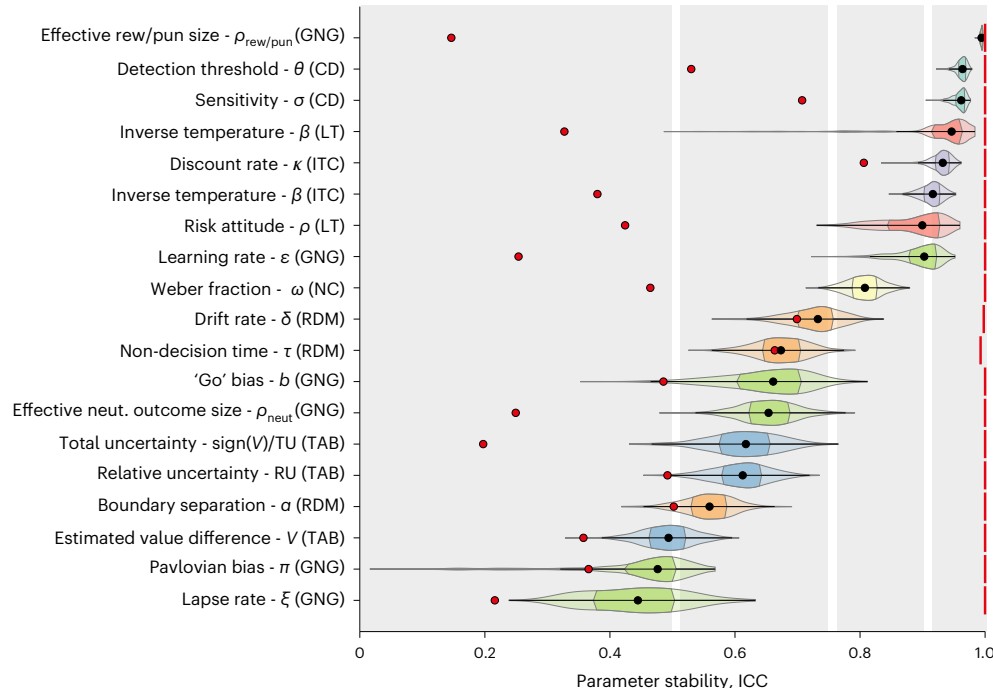

**Fig. 2 | Stability of the computational phenotype.** Violin plots show the bootstrapped stability estimates in terms of ICC for each parameter in the computational phenotype (1,000 bootstrap iterations over participants; violin colours indicate tasks). Black points mark the median, colour shaded areas mark the interquartile range (IQR). Red vertical lines mark the upper stability bound estimated using simulated data with a phenotype that is fixed in time. Red points mark the median ICC values estimated using the reduced statistical model, which does not account for the full hierarchical structure of the data. Note that most

of the reduced model estimates fall within the area indicating poor reliability (see Supplementary Information for more details and discussion). Grey regions mark a common interpretation of ICC values[35]: <0.5 (poor), 0.5–0.75 (moderate), 0.75–0.9 (good) and >0.9 (excellent). ICC was calculated only in participants without missing sessions: Go/No-go (GNG): 53 participants out of 66 (53/66), Change detection (CD): 65/90, Intertemporal choice (ITC): 67/90, Lottery ticket (LT): 47/58, Numerosity comparison (NC): 69/90, Two-armed bandit (TAB): 66/90, Random dot motion (RDM): 73/90.

---

task has the richest computational phenotype (six parameters), and it illustrates all of the dynamic effects (practice and mood) found in other tasks. We then summarize the results of the remaining six tasks (see Supplementary Information and Supplementary Figs. 6 and 7 for further details). Finally, across all tasks we verified that the reported practice effects persist also after excluding the first session, meaning that they are not an artefact of insufficient understanding of the task at the first session (see Supplementary Information).

**Structure and noise in the temporal variability of the Go/No-go task**

In the Go/No-go task, participants were trained to choose between pressing a key on their keyboard ('Go') or withholding a key press ('No-go') depending on which of four visual stimuli they were shown. Participants were probabilistically rewarded or punished on the basis of their responses. Previous work[36] developed a model that captures the key behavioural characteristics in this task, which we adopt in our study. This model posits that responses depend on three factors: expected reward for each stimulus–response combination, an unlearned 'Go' bias and a learned 'Go' bias that increases with average reward for a cue and decreases with average punishment. The learned bias allows the model to capture the tendency to approach rewarding stimuli ('Go') and avoid punishing stimuli ('No-go'), a form of Pavlovian misbehaviour that can disrupt correct instrumental performance. This manifests as an increased probability of incorrect 'Go' responses in the 'No-go to win' condition and an increased probability of incorrect 'No-go' responses in the 'Go to avoid' condition. Notably, we departed from the original model[36] by modelling neutral outcomes as rewards (+1) in the punishment conditions and as punishments (−1) in the reward conditions. This reflects the assumption that participants perceive outcomes in a context-dependent manner[37,38].

The strengths of the 'Go' biases are controlled by two parameters: $b$ for the unlearned (instrumental) bias and $\pi$ for the learned (Pavlovian) bias. The instrumental and Pavlovian learning processes are jointly controlled by a learning rate parameter ($\varepsilon$) and the 'effective size' ($\rho$) of different outcome types. We fit a separate $\rho$ parameter for neutral outcomes to accommodate their distinct role in governing participants' performance. Finally, noise in the action selection process is controlled by a lapse rate parameter ($\xi$) that captures the probability of random responses.

At the behavioural level, participants' performance improved across the 12 weeks. Figure 3a shows the group-level overall accuracy and accuracy in each task condition over time. All conditions except 'Go to win' were well fit by a power-law curve ($R^2_{adj} > 0.8$, $P < 0.001$, random permutation test of the session order), consistent with classic models of practice effects[39,40]. Turning to the independent model fits, the posterior predictive checks indicate that our version of this Pavlovian reinforcement learning (RL) model, which treats neutral outcomes as potential reinforcers, captures the key behavioural characteristics of the task (Fig. 3b; see Supplementary Figs. 9 and 10 for the results of the original model proposed in ref. 36). Figure 4 shows the mean estimated parameters over time in all participants. Some parameters (for example, the effective neutral outcome size) show a clear trend over time, while others show fluctuations with no clear structure (for example, the lapse rate). Parameter stability for this task spanned a wide range: lapse rate $\xi$ (median ICC = 0.45), Pavlovian bias $\pi$ (0.48), effective size of neutral outcomes $\rho_{neut}$ (0.65), Go bias $b$ (0.66), learning rate $\varepsilon$ (0.9) and effective size of reward/punishment $\rho_{rew\backslash pun}$ (0.99).

To better understand potential sources of temporal variation, we refit the Pavlovian RL model with a dynamic model in which parameter values are governed by three sources of temporal variation: (1) practice effects (that is, monotonic directional changes); (2) state effects

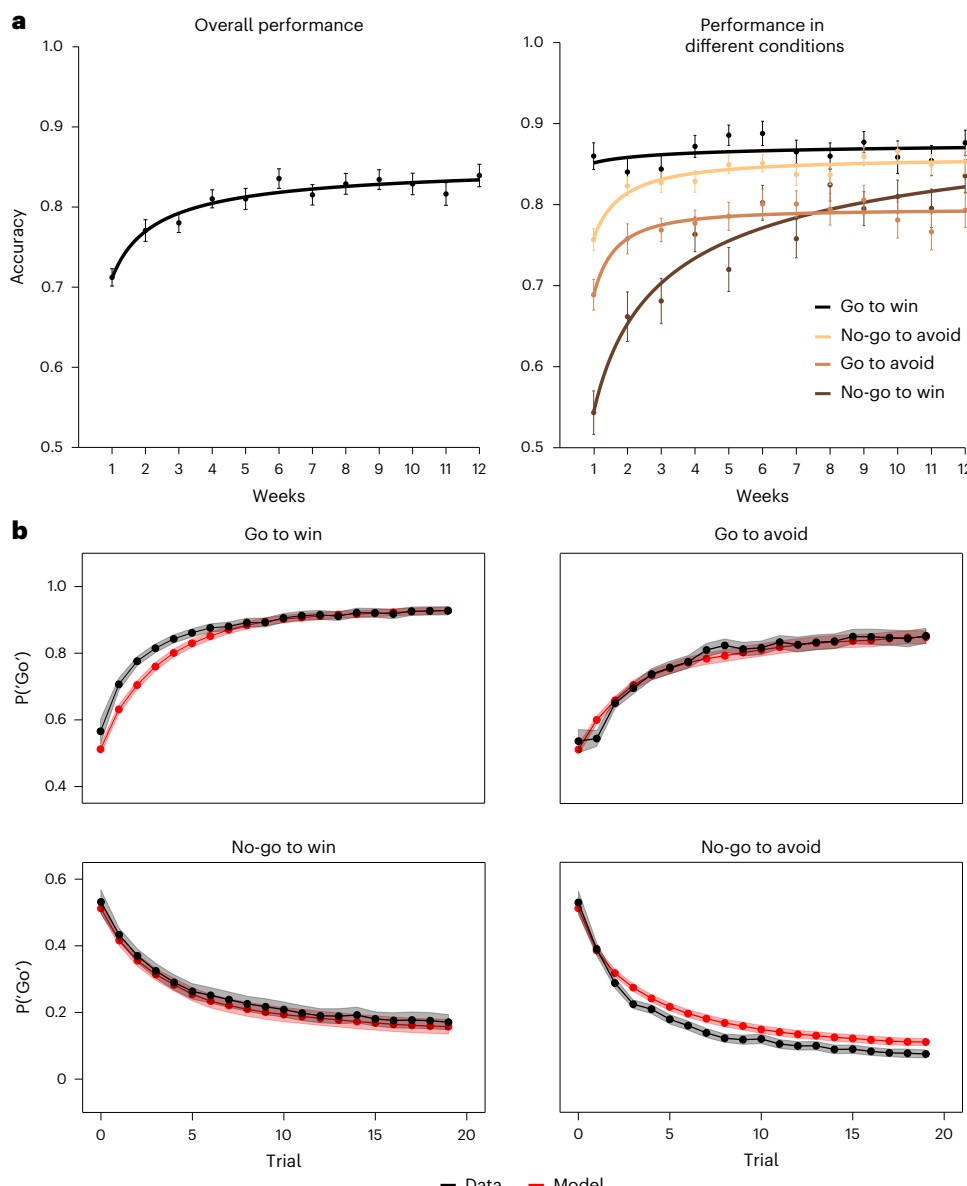

**Fig. 3 | Behavioural performance and posterior predictive checks in the Go/No-go task. a**, Accuracy over time. Both overall accuracy (left) and accuracy in different task conditions (right) increased over time (except for the 'Go to win' condition, which was already at ceiling). We fit the data with a power-law curve and report the $P$ value of $R^2_{adj}$ based on 10,000 random permutations of the session order. Error bars show the s.e.m. across participants. **b**, Posterior predictive checks. We used the parameter estimates from the independent model to predict the probability of taking a 'Go' action in different task conditions. Model fits (red) closely matched behaviour (black). Real and predicted data were averaged across blocks, sessions and participants. Predicted data were also averaged across samples of the posterior distribution. Shaded areas show the s.e.m. across participants.

(affective valence and arousal, extracted from an analysis of the survey data; Supplementary Fig. 12); and (3) random noise. Practice effects were modelled as a power-law change in time[39,40], while state effects were modelled as fluctuations around each participant's baseline, based on the daily self-report surveys. Intuitively, valence captures the positivity versus negativity axis of participants' affective state, whereas arousal captures the excited versus calm axis[41]. Finally, we added a random noise effect to model any other source of variability that is not accounted for by the aforementioned sources. We quantified the relative contribution (RC) of each source to the computational phenotype by calculating its standard deviation over time relative to the sum of standard deviations of all the sources. Figure 4 shows the relative contributions of different sources to each of the phenotype parameters in the Go/No-go task. Noise had the strongest contribution for 4 of 6 task parameters, followed by practice effects. See Supplementary Fig. 2

for a breakdown of the conventional variance components underlying the ICC calculation.

To quantify the statistical strength of each source contribution, we calculated the probability of direction (PD), that is, the proportion of the posterior distribution that has the same sign as the median[42]. Rather than a binary cut-off such as exclusion of zero from the highest density interval, this allows for a graded quantification of the effects, with recommended interpretation of ≤95 ('uncertain'), ≥95 ('possibly existing'), ≥97 ('likely existing'), ≥99 ('probably existing') and ≥99.9 ('certainly existing'). For the Pavlovian bias, we found a certainly existing negative effect of practice (PD = 100, median RC = 0.23). The sign of the reported effect (positive or negative) indicates the direction of the association between each parameter and the variability source (indicated by black arrows in Fig. 4; for example, the Pavlovian bias decreased with practice). For the lapse rate, valence had a probably

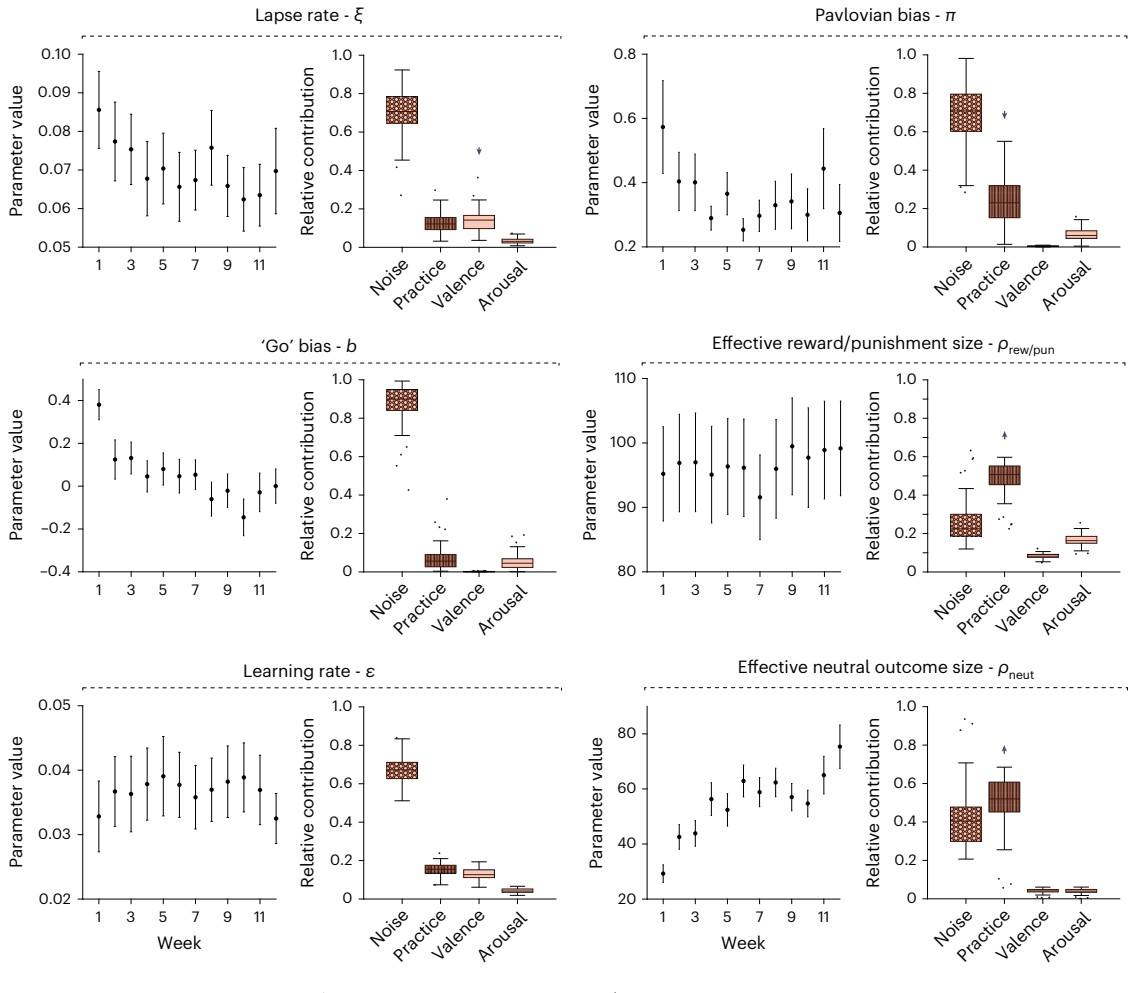

↑ Increase in paramater, PD > 95   ↓ Decrease in paramater, PD > 95

**Fig. 4 | The dynamic computational phenotype in the Go/No-go task.** For each parameter, the left plot shows its mean ± s.e.m. across participants in each week (derived from the independent model; error bars show the s.e.m. across participants), and the right plot shows the relative contribution of each source (derived from the dynamic model). Arrows indicate the direction of the effect on each parameter over time for effects with PD > 95: up, positive; down, negative. The centre line marks the median, the box limits indicate the IQR and whiskers extend to the most extreme data points not considered outliers (points that are farther than 1.5 IQR from the box limits).

existing negative effect of practice (PD = 97, median RC = 0.14). For the effective size of reward/punishment, as well as the effective size of neutral outcomes, we found certainly existing positive effects of practice (PD = 100, median RC = 0.51 and 0.52, respectively).

These results indicate heterogeneous sources of variability across the computational phenotype in the Go/No-go task, where some parameters are driven mostly by practice effects ($\rho_{rew\backslash pun}$, $\rho_{neut}$, $\pi$), while others are driven mostly by random noise ($\xi$, $b$, $\varepsilon$). See Supplementary Information for similar analysis of the 'non-learning' participants which were excluded due to low task performance[36,43].

Below we report the results for each one of the remaining tasks. For each task, we give a brief description of the task followed by the relevant computational model, and then report the stability scores of the phenotype parameters and the dynamic effects that account for their temporal variability. The relative contribution of each source to the computational phenotype of these tasks is presented in Fig. 5.

### Change detection

In the Change detection task, participants had to indicate whether two consecutively presented images depicting coloured squares were identical. We modelled this task using the absolute difference model (MAD)[44], which assumes that participants compare each pair of corresponding squares in the two images independently on the basis of

noise-corrupted representations. Participants deem the corresponding squares different with a probability that depends on a detection threshold and on the absolute difference between their representations. The noise in the image representation is accounted for by a sensitivity parameter $\sigma$, and the decision threshold is controlled by the detection threshold $\theta$. Here we assume that both parameters increase linearly with set size[44]. This requires two more free parameters: the slope of $\sigma$ and the slope of $\theta$ with set size. Here we focus on the dynamics of $\sigma$ and $\theta$.

At the behavioural level, most task conditions showed stable accuracy values over time, except for the eight-items-single-target condition (Supplementary Fig. 6). Accuracy in this condition was well captured by a power-law curve ($R^2_{adj} = 0.7$, $P < 0.01$, random permutation test). The detection threshold and noise parameters were highly stable (median ICC = 0.96; Fig. 2). The dynamic model revealed certainly existing positive effects of practice for detection threshold (PD = 100, median RC = 0.37) and for sensitivity (PD = 100, median RC = 0.33; Fig. 5).

### Random dot motion

In the Random dot motion task, participants viewed 200 moving dots and had to indicate the direction (left/right) of the coherently moving dots among them. Coherence levels were 5%, 10%, 35% or 50% of all dots.

To model participants' responses and reaction times, we used the drift diffusion model[45,46]. This model assumes that participants

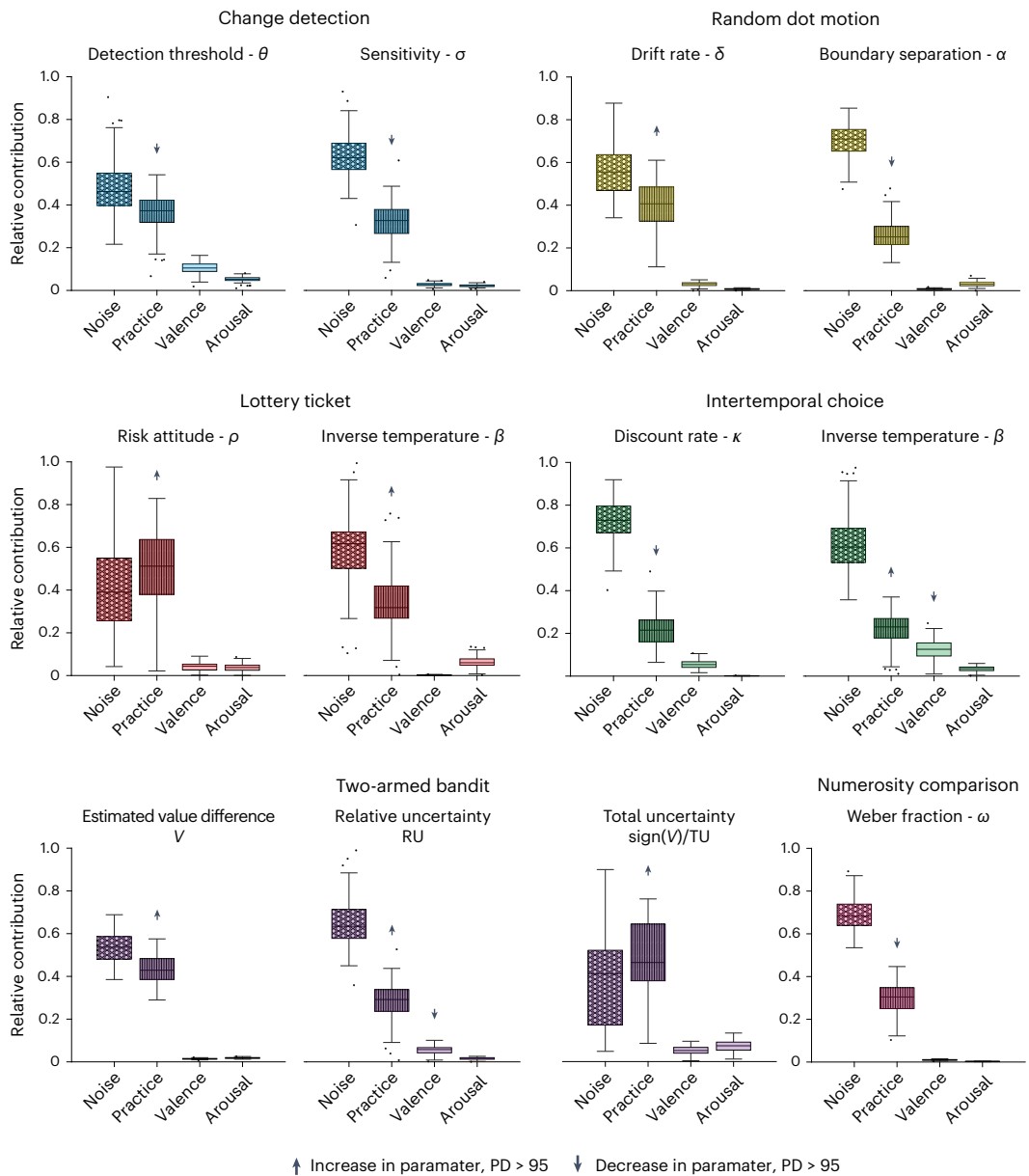

**Fig. 5 | Sources of temporal variation across tasks.** For each parameter, the relative contribution of sources (derived from the dynamic model) are shown, organized by task. Arrows indicate the direction of the effect on each parameter over time for effects with PD > 95: up, positive; down, negative. The centre line, box limits and whiskers are interpreted as in Fig. 4.

accumulate noisy evidence in favour of each potential response and respond when the accumulated evidence passes a certain threshold.

In the drift diffusion model, evidence accumulation is controlled by the drift rate parameter $\delta$. Here we assume that the drift rate is linear with the coherence level[47] and thus fit a single drift rate for each session. The threshold for decision is controlled by the decision boundary $\alpha$. Lastly, the reaction times are modelled as the sum of the evidence accumulation time and the non-decision time $\tau$, which captures processes such as stimulus encoding and motor response.

Behaviourally, the accuracy across all motion coherence values improved over time and all conditions were well described by a power-law curve ($R^2_{adj} \geq 0.9$, $P < 0.01$ for all conditions using a random permutation test; Supplementary Fig. 6). Model parameters showed moderate stability: decision boundary $\alpha$ (median ICC = 0.56), non-decision time $\tau$ (0.68) and drift rate $\delta$ (0.73; Fig. 2). Supplementary Fig. 7 shows the mean parameter values over time estimated using the independent model. While the decision boundary

$\alpha$ and the non-decision time $\tau$ decreased, the drift rate $\delta$ increased with practice. Using the dynamic model, we found a certainly existing negative effect of practice (PD = 100, median RC = 0.25) for the decision boundary and a certainly existing positive effect of practice (PD = 100, median RC = 0.41) for the drift rate (Fig. 5).

**Lottery ticket**

In the Lottery ticket task, participants chose to bet on one of two displayed tickets[48]. The 'risky' ticket offered a gamble between a very high and a very low sum of money, while the 'safe' ticket offered a gamble between two moderate sums of money. We used a risk-sensitive utility function[48] to estimate the perceived expected value of each ticket, which was governed by the risk attitude parameter $\rho$. Higher $\rho$ indicates a higher tendency to prefer risky choices. We used a soft-max probability function to model participants' choice stochasticity, governed by the inverse temperature parameter $\beta$. Larger $\beta$ indicates less stochastic behaviour.

Both task parameters were highly stable (median ICC = 0.95 and 0.9, respectively; Fig. 2). Supplementary Fig. 7 shows the mean parameter values over time estimated using the independent model. Using the dynamic model, we found a certainly existing positive practice effect (PD = 100, median RC = 0.32) for inverse temperature. For the risk attitude, we found a certainly existing positive effect of practice (PD = 100, median RC = 0.51).

See Supplementary Information for similar analysis of the excluded participants who chose the 'safe' ticket almost exclusively. Note that attempting to fit the data of all participants simultaneously resulted in severe convergence issues, indicated by large R-hat values and by a large number of divergences. This is probably the result of the extreme differences in phenotype parameters that best explain the different behaviour patterns of the two groups (Supplementary Figs. 16 and 17).

### Intertemporal choice
In the Intertemporal choice task, participants were asked to choose between a small but immediate amount of money and a larger but temporally delayed amount.

For modelling participants' behaviour, we adopted a previously introduced model[49] in which the tendency to prefer delayed rewards is represented by the discount rate parameter $k$. Larger $k$ values indicate greater impulsivity and preference to receive immediate rewards. We adapted the original hyperbolic function model with a softmax probability function to model participants' choice stochasticity, governed by the inverse temperature parameter $\beta$[50]. See Supplementary Information and Supplementary Fig. 18 for an analysis based on the original model without the softmax decision rule.

Both task parameters showed excellent stability (median ICC = 0.91 and 0.93 for $\beta$ and $k$, respectively; Fig. 2). Supplementary Fig. 7 shows mean parameter values over time based on the independent model, suggesting that inverse temperature increases with practice. Using the dynamic model, we found a certainly existing effect of practice on the inverse temperature (PD = 100, median RC = 0.23), as well as a certainly existing effect of valence (PD = 100, median RC = 0.13). These results indicate that decisions became more deterministic with practice. For the discount rate, we found a certainly existing effect of practice (PD = 100, median RC = 0.21), associated with reduced discounting of future reward.

### Two-armed bandit
In the Two-armed bandit task, participants chose between two 'slot machines' labelled either R (risky) or S (safe), yielding four conditions, which we refer to as SS, RR, RS and SR based on the presented labels. While risky machines led to a different reward outcome every time, the safe machines provided the same outcome every time. To model participants' behaviour, we used a previously introduced model[51], which accounts for different types of exploratory behaviour. Specifically, we used a probit regression model in which choice is probabilistically governed by three factors. Exploitative choices are promoted by the weight of the estimated value difference of the two machines $w_V$. Directed exploration is governed by the weight of the relative uncertainty between the two machines $w_{RU}$. Random exploration is controlled by the weight of the signed total uncertainty $w_{\text{sign}(V)/\text{TU}}$. We deviated from the original model by using the sign of $V$ rather than $V$ itself. This promoted model convergence and led to a better fit to behavioural data in an independent dataset (see Supplementary Information).

Behaviourally, the probability of choosing the machine with the higher expected value increases over weeks in all conditions and is fit reasonably well by a power-law curve (Supplementary Fig. 6; RS/SR: $R^2_{\text{adj}} \geq 0.4, P < 0.05$, random permutation test; SS: $R^2_{\text{adj}} = 0.8, P < 0.01$; RR: $R^2_{\text{adj}} = 0.6, P < 0.01$, random permutation test). Using the independent model, we found that all task parameters increased with practice (Supplementary Fig. 7). While the weight for the estimated value difference $w_V$ seemed to increase gradually, the weights for

uncertainty-driven behaviour ($w_{\text{sign}(V)/\text{TU}}$ and $w_{RU}$) dramatically increased between the first and second sessions, and fluctuated afterwards.

These observations are in line with the stability results for the model parameters (Fig. 2). The weight for estimated value difference between arms $w_V$ showed poor stability (median ICC = 0.49), while the weights for relative uncertainty $w_{RU}$ and signed total uncertainty $w_{\text{sign}(V)/\text{TU}}$ showed moderate stability (median ICC = 0.61 and 0.62, respectively). Using the dynamic model, we found a certainly existing positive effect of practice for all model parameters (Fig. 5; PD = 100, median RC = 0.43 for the $w_V$, median RC = 0.48 for $w_{\text{sign}(V)/\text{TU}}$ and median RC = 0.29 for $w_{RU}$). For the weight of relative uncertainty $w_{RU}$, we also found a likely existing effect of valence (PD = 98, median RC = 0.058).

### Numerosity comparison
In the Numerosity comparison task, participants indicated which of two presented clusters of golden coins contained more coins. To model participants' responses, we assumed that the number of coins in each cluster is encoded as a noisy estimate that follows a normal distribution centred around the true value $a$, with a magnitude-dependent noise $a^2w^2$ (ref. 52). The parameter $w$ is the Weber fraction, which controls the magnitude-dependence of noise.

Behaviourally, participants' accuracy in this task was very stable (Supplementary Fig. 6). The subtle improvement in accuracy over time was fit well by a power-law curve ($R^2_{\text{adj}} = 0.65, P < 0.001$, random permutation test). Inspection of the average parameter value based on the baseline model did not indicate any clear temporal trend. The Weber fraction showed good stability (median ICC = 0.81). Using the dynamic model, we found a certainly existing negative effect of practice (PD = 100, median RC = 0.3).

## Discussion
In this study, we introduced a longitudinal dataset and a statistical framework that together allowed us to explore the dynamic nature of the computational phenotype. Our model, fit to 90 participants performing 7 tasks across 12 weeks, revealed the existence of three dynamical processes influencing computational parameters: practice, affective state and random noise. While this list is not exhaustive, it supports the view that the nature of the computational phenotype is dynamic and that some of its variability tracks meaningful changes in participant-related factors, rather than simply reflecting unreliability (cf. 24,53). This view emphasizes the structured temporal variation in the computational phenotype and suggests that it should be measured to provide insight into inter- and intra-individual cognitive variation.

### Practice effects
In our study, almost all of the parameters comprising the computational phenotype were affected by practice. Even though we aimed to reduce practice effects by using different stimuli across weeks (for example, different fractal images in the Go/No-go task), we found a monotonic increase in the average accuracy in tasks where objective accuracy can be defined. Interestingly, in tasks that indicate personal preference, such as the Intertemporal choice task, the choices that participants made became less stochastic with practice.

Our findings are consistent with existing studies examining the effects of repetitive task administration on performance and on the estimated computational phenotype. The behavioural practice effects associated with improved task performance we report in our study are consistent with previous studies on repetitive administration of the Random dot motion[23,54–56], Numerosity comparison[57] and Change detection[58] tasks. The changes we observed in the parameters of the models for the Random dot motion and the Intertemporal choice tasks are also consistent with the previous literature[23,56,59].

Even though task performance was already adequate in the first session, we observed continued improvement in task performance for some of the tasks well after the first few sessions. Since practice

effects are a main confounding factor in longitudinal studies, these results imply that one cannot safely assume that task performance plateaus after a session or two, let alone after baseline practice (see discussion in ref. 20). Nevertheless, we found that practice effects were captured well by a power-law function, meaning that the relative contribution of practice effects diminishes with time. This is important for longitudinal experimental designs, since it could potentially allow other meaningful sources of longitudinal variability to become more prominent in later sessions (when they are less likely to be washed out by large practice effects).

### Affective state effects

Previous studies examined the effects of affect on the computational phenotype by manipulating mood or stress. For example, acute stress induction altered 'model-based' but not 'model-free' decision making[60]. In another study, induction of positive or negative mood throughout the task was associated with biased reward perception[61,62]. In the current study, we focused on day-to-day spontaneous mood fluctuations[63]. We found that these fluctuations (particularly affective valence) were associated with temporal variation in 3 parameters: inverse temperature (Intertemporal choice), lapse rate (Go/No-go) and weight of relative uncertainty (Two-armed bandit). Interestingly, these parameters are all associated with choice stochasticity. However, the specific effect of valence on choice stochasticity differed between parameters. Higher valence (more positive mood) was associated with more stochastic choices in the Intertemporal choice task (lower inverse temperature) but more deterministic choices in the Go/No-go task (lower lapse rate). In the Two-armed bandit task, higher valence was associated with lower weight of relative uncertainty, thereby reducing the effect of uncertainty-guided directed exploration.

These findings hint at a complex relationship between mood and choice stochasticity. The positive relationship between valence and decreased choice stochasticity we report for two of three parameters is partially consistent with a previous study showing that positive mood predicted increased exploitation and high arousal predicted increased exploration[64] (but see ref. 65). Our mixed findings could reflect fundamentally different sources of choice stochasticity in different tasks, each affected differently by mood. Our data-driven approach using principal component analysis (PCA) led to a general valence component, and future studies might benefit from a fine-grained analysis of the specific effects of different emotions on the computational phenotype.

### A longitudinal design improves our understanding of cognition

Our findings show that employing a longitudinal design permits a more nuanced comprehension of cognitive processes across longer time-scales. An illustrative case of this point is the Go/No-go task. Using a well-established Pavlovian RL model of this task[36], we were not able to fully capture participants' behaviour (Supplementary Fig. 9). Specifically, the model failed to account for participants' correct responses in the avoid-punishment conditions. Previous work using other reinforcement learning tasks suggests that participants perceive outcomes in a context-dependent manner[37,38]. On the basis of this idea, we were able to resolve the disagreement between the predicted and observed data by introducing a minor modification to the model, allowing the agent to learn from neutral outcomes (see Methods). Specifically, the modified model assumes that participants perceive neutral stimuli as rewards in the presence of punishments and as punishments in the presence of rewards. It is noteworthy that the original model adequately fit the data from the first session (Supplementary Fig. 10), which may explain why the proposed modified model remained undiscovered until a longitudinal perspective was taken. Only through observing participants' behaviour over time did this subtle nuance emerge as a robust effect.

One additional striking observation which became evident with the longitudinal design is the increased accuracy in the 'No-go to win' condition. One of the most established behavioural findings in the Go/No-go paradigm is that participants are better in learning 'Go' responses in reward conditions and 'No-Go' in punishment conditions[36]. In our study, while we observed improvement in all conditions of the Go/No-go task over sessions, the 'No-go to win' condition showed the most significant improvement in accuracy and even surpassed the 'Go to avoid' condition, thus reducing the preference towards 'No-go' responses in the punishment conditions. This is probably explained by the decrease we found in the Pavlovian bias with practice, such that participants became less likely to withhold a key press in this punishment condition.

### Modelling limitations

For most of the parameters in the computational phenotype, a large fraction of the temporal variability was not attributed to one of the systematic sources we explored in this study (practice and state effects) but was associated with noise. One reason for this could be lack of relevant data that could have been collected longitudinally. While the daily surveys covered a wide range of questions that targeted participants' mood and daily habits, it is possible that relevant factors remained unexplored. Furthermore, the surveys related to the participants' daily state, rather than to momentary state during task performance (for example, ref. 61, which would require a more frequent and fine-grain mood assessment), which was out of the scope of the current study.

Alternatively, it might be possible to reduce the prominence of noise by refining the dynamic model. For simplicity, as well as for promoting the convergence of our estimation procedure, we assumed similar effects across participants. In modelling the practice effects, we assumed a power-law change over time, whereby the magnitude is modulated by the participant-specific baseline, and the exponent was sampled from a population distribution. In modelling the state effects, we normalized the state measures within each participant and assumed population-level effects only. We made these modelling choices to tractably capture the primary effects observed in the data. This approach may conceal more nuanced participant-specific effects that potentially await to be revealed. To capture such effects, a more intricate modelling approach could be pursued, albeit at the cost of increased model complexity and potential challenges in parameter estimation.

Overall, we believe that capturing additional sources of variability in the computational phenotype is necessary to gain a deeper understanding of the underlying cognitive processes. While understanding these sources will not necessarily make the measurement more stable, in principle it could be used to obtain a 'cleaner' measurement. For example, by collecting data that account for participants' affective state, one could regress out the state effects from the measured phenotype. Structural and functional neural states could also be longitudinally recorded concurrently with task performance to increase modelling accuracy in future studies[66]. Neural data can also be used to associate the computational phenotype with specific biological phenotypes[67]. Although collecting reliable neurophysiological daily point-estimates is a challenging task, recent advances in individualized brain imaging make this direction practically feasible[68–72] and particularly exciting in the context of translational neuroscience, potential for clinical application and precision medicine[73,74].

It is important to consider that participants in our study performed each task once a week, meaning that our dynamic model captured practice-related and affect-related effects on the basis of this temporal resolution of longitudinal probing. The effects we report might be different when applying a different experimental protocol where participants perform the tasks every day or once a month. For example, we would expect the practice effects to be reduced as the spacing between sessions is increased.

The results we present here in terms of phenotype dynamics over time could depend on the specific variant of the task. For example, different methods for risk elicitation in the Lottery ticket task differ in

terms of parameter magnitude and reliability[75,76]. More work is needed to determine how parameter stability is affected by the presentation method (for example, presenting the different lotteries as a single list or trial by trial, as we did here).

Different statistical approaches could be used to study the effects that influence the dynamics of the computational phenotype. We chose a generative Bayesian model that makes explicit assumptions about the functional form of these dynamical effects and their underlying prior distributions. The main advantages of this approach are in providing the full posterior distribution of the model parameters and in providing a suitably flexible hypothesis space supporting stronger inference. This contrasts with techniques such as linear regression, which is less suitable in this case, since its hypothesis space is too restrictive. Consider for example the well-established power-law form of practice effects[39,40]. While our generative model allows fitting of power-law parameters for each participant, a linear regression model would require defining a fixed practice curve across the population, with the regression coefficient as a single free parameter that scales this curve.

An interesting future direction would be to model all tasks simultaneously, rather than one by one. This would allow revelation of potential relationships in the computational phenotype across tasks and uncovering of unified cognitive processes that underlie seemingly disparate computational parameters. Our findings reveal one challenge associated with this approach: the prevalence of practice effects might lead to spurious correlations among computational parameters that are in fact independent of each other. Therefore, a systematic examination of cross-task relationships might require a more sophisticated experimental design that would allow us to tease apart spurious and true sources of shared variation.

The modelling framework we present here combines task-specific computational models with a general dynamic model of the computational phenotype. Together, they comprise a generative model in the sense that given model parameters, one can produce a time series of phenotype values (with practice effects and affective effects governed by the model parameters) and simulate behavioural data[77]. This generative model, however, is not mechanistic[78]. While we hint at specific relations between certain state effects (for example, affective valence) and phenotype parameters (for example, uncertainty-guided directed exploration), understanding a mechanism that underlies these relationships is an open question in cognitive neuroscience.

Establishing the reliability of the computational phenotype is crucial if one seeks to study human cognition through computational models of behaviour. As illustrated in ref. 20, characterizing parameter reliability is only the first step in a hierarchy of steps, which was beyond the scope of our current research. An important next step in laying the psychometric foundation of computational phenotypes would be to establish their external validity and to compare it with the external validity of simpler summary statistics of task performance[24].

## Conclusions

Our study demonstrates that the computational phenotype can be highly dynamic. These dynamics to some extent reflect random fluctuations and in this sense are truly 'unreliable' (that is, there is temporal variation that we do not yet know how to capture). At the same time, many parameters show structured covariation with time (specifically, practice effects) and with affective state. This suggests that the quest for reliable computational phenotypes may be the wrong way to think about such constructs. Rather, the quest must be redirected towards understanding the multiple sources of temporal variation underlying cognition.

## Methods

### Participants

Participants were recruited from Amazon Mechanical Turk through CloudResearch services. To increase the likelihood of continued participation throughout the study, the participant pool was filtered to have an approval rating of 90% or above. Participants were compensated at a rate of US$10 per hour (henceforth, $ are in US$). An additional payment of $6 per week in task bonuses could also be earned on the basis of performance. Participants were also eligible to receive one-time $5 and $10 bonuses on the basis of continued participation after 4 weeks and 8 weeks, respectively (study milestones). Participants who finished all 12 weeks of the study were awarded an additional $15 completion bonus. Participants read the study description via Amazon Mechanical Turk and indicated whether they agree to participate with a button press. The study was approved by the Institutional Review Board of Harvard University and was performed in accordance with the relevant guidelines and regulations.

In total, 141 participants were recruited for the study. Participants were considered to take part in the study if they provided answers to the demographics and personal survey questionnaires that were administered together on the first day of the study. Due to a technical issue, demographic data were not collected from one participant. Since the main interest of the study was to examine the temporal dynamics of the computational phenotype, we only included participants who had at least six weeks of task data with no more than three missing consecutive weeks for each task. Out of the initial cohort, 90 participants (mean age: 39.4 ± 10.8 years; 47 identified as males, 41 identified as females, 1 identified as queer) satisfied these criteria. All further analyses were performed on this subset of participants.

### Design and procedure

The study comprised seven experimental tasks (described below). On the basis of the average amount of time required for each task, the tasks were divided into three groups or 'sessions' performed on a weekly basis (two to three tasks per day). The order of tasks was shuffled each week within the session; however, tasks did not rotate between sessions and session order remained fixed for all participants throughout the entire study. Participants completed 1 session a day for 3 consecutive days, every week for 12 consecutive weeks. At the beginning of each session, participants completed a survey that asked 36 questions pertaining to their mood and health (for example, "How stressed are you now?"; see Supplementary Information for all the survey questions). In the first session of the experiment, participants also completed a survey about their demographic data and personality traits, including the Barratt Impulsiveness Scale[79] and the DOSPERT scale for domain-specific risk attitude[80].

Below we summarize the procedure for each task. Detailed descriptions can be found in Supplementary Information.

**Go/No-go.** The design of this task was modified from a previously published version[36]. The task had 3 blocks, each 80 trials long with a distinct set of 4 visual stimuli. On each trial, participants were shown one of the stimuli and asked to choose between pressing the space bar ('Go') or withholding a press ('No-go'). At the end of the trial, participants received feedback in the form of reward (gain of points), punishment (loss of points) or a neutral outcome (neither gain or loss of points). Each stimulus prescribed a correct response: 'Go to win', 'Go to avoid punishment', 'No-go to win' and 'No-go to avoid punishment'. If a stimulus was associated with the outcome rule 'to win' and the correct response was selected, there was an 80% chance that the participant would win points (indicated by a green thumbs-up image) and a 20% chance that they would neither win nor lose points (indicated by a flat grey hand image). If the stimulus was associated with the outcome rule 'to avoid punishment' and the correct response was selected, there was an 80% chance that the participant would neither win nor lose points (indicated by a flat grey hand image) and a 20% chance that they would lose points (indicated by a red thumbs-down image). If an incorrect response was selected, there was an 80% chance of receiving a neutral outcome in the win

conditions or a punishment outcome in the punishment conditions. Each trial was followed by a 750–1,500 ms intertrial interval with a fixation cross.

Following previous work[36], we classified participants into learners and non-learners. We defined non-learners as participants who had an average accuracy of less than 0.55 throughout the study. We focused our analysis on the 66 participants who were classified as learners out of 90 participants (for comparison, a previous study[36] found a similar fraction of learners, 19 out of 30 participants).

**Change detection.** The Change detection task, based on previous work[44], tests the participants' ability to detect a change between pairs of images. The images were sets of coloured squares arranged in a ring formation. On each trial, one image was shown for 100 ms, followed by an intertrial interval (grey screen with white fixation cross presented for 1,500 ms), and then a probe image, which was either the same as the first image (no-change condition) or different (change condition). In the change condition, one or more of the squares change colour. After the second image was shown, participants were asked whether the two presented images were the same or different and how confident they were in their decision. Participants received feedback stating whether they were correct or incorrect.

The task consisted of 5 blocks, each with 40 trials. In four of the blocks, the set size (the number of squares in the array) was either three, four, six or eight squares. In addition, there was one block with a set size of eight squares, in which between zero and four squares changed colour on each trial (the 'multiple targets' block). Set size did not change within blocks except during the practice trials.

**Random dot motion.** On each trial of the Random dot motion task, participants were tasked with reporting the direction (left versus right) of a cloud of moving dots. While some percentage of the dots (5%, 10%, 35% or 50%) were moving coherently in the same direction, other dots were moving in random directions. Participants completed 4 experimental blocks, each containing 96 trials. On every trial, the participant was shown a cluster of 200 randomly distributed white dots on a dark grey background. Participants had 1,500 ms to determine the direction of motion of the coherently moving dots. After each trial, a 300 ms intertrial interval showing a centred white fixation cross on a dark grey background was shown.

**Lottery ticket.** In the Lottery ticket task, participants chose to bet on one of two possible tickets[48]. The 'risky' ticket offered a gamble between a very high and a very low sum of money. The 'safe' ticket offered a gamble between two moderate sums of money. The odds associated with the outcomes were always the same between tickets. For example, the safe ticket might provide a 20% chance of winning $2 and an 80% chance of winning $1.60, while the risky ticket provided a 20% chance of winning $3.85 and an 80% chance of winning $0.10. The task consisted of 3 blocks, each 10 trials long. Each block utilized a different monetary range: low (tickets with expected value (EV) of approximately $2), medium (EV ≈ $50) and high (EV ≈ $200). For modelling purposes, we removed trials with a win probability of 100%.

Even though this task reflects personal preference and there are no objective correct or incorrect decisions, we excluded participants who chose the same option on over 80% of the trials (these were all participants who chose the 'Safe' option), leaving 58 participants for further analyses pertaining to this task.

**Intertemporal choice.** The Intertemporal choice task is a subjective monetary questionnaire in which participants indicated their preference between a smaller amount of money available immediately or a larger amount of money available at a later time. This task is based on previous work[49]. Each delayed monetary reward (small, medium, large) was used three times in the task, for a total of 27 trials.

**Two-armed bandit.** This task closely followed a previously published design[51]. On each trial, participants chose between two 'slot machines' labelled either R (risky) or S (safe). Risky machines led to a different reward outcome every time; safe machines provided the same outcome every time. Participants were informed that one machine was always better than the other and their task was to make choices that earn as much reward as possible. Participants had to explore each machine to determine which one had the higher expected reward. The task consisted of 30 blocks, each 10 trials long. The mean and variance of the machines changed across blocks.

**Numerosity comparison.** In the Numerosity comparison task, participants were required to make a judgement about which 'treasure chest' has more 'gold coins' hovering above it. Participants were told that all gold coins have the same value regardless of size. Gold coins were represented by yellow dots on a black background in an invisibly bounded section above a treasure chest. There were always two chests presented on a screen in every trial, with one chest being on the left side of the screen and the other being on the right side[31]. The task was 160 trials long. No two chests had exactly the same number of gold coins.

### Personal survey
The personal survey consisted of 36 questions. We excluded one question that allowed participants to freely express any issue they find relevant that was not captured by the rest of the questions (question no. 34 in the survey, see Supplementary Methods). To reduce the number of items used in the dynamic model (see below), we ran a dimensionality reduction analysis on the survey responses. For each survey item, we z-scored the responses across participants (ignoring missing sessions) and then performed PCA. We used the top two components, which explained 32% and 9% of the variance in the data, respectively. Missing sessions were linearly interpolated after running PCA. On the basis of the principal component loadings, we interpreted these components as follows (Supplementary Fig. 12):

- PC1 - Affective valence: this PC loaded positively on items assessing positive emotions (for example, enthusiasm, happiness, relaxation) and negatively on items assessing negative emotions (for example, nervousness, being upset, irritation, sadness, loneliness, hostility).
- PC2 - Affective arousal: this PC loaded positively on all items assessing affective state, as well as on items assessing stress, energy levels, being active, sleeping well and eating healthy.

### Statistical models
We modelled all the task data using a hierarchical Bayesian framework, which has been shown to improve the accuracy of parameter estimation compared with methods that fit the data of each participant (or session) independently[32,33]. Specifically, for each task, the parameters for a given session were drawn from a participant-specific prior distribution. The parameters of this prior distribution, in turn, were drawn from a group-level distribution with weakly informative hyper-priors. As detailed below, we fit two main variants of this hierarchical model: an independent model and a dynamic model. The key difference between them is that the independent model assumes that session-level parameters are conditionally independent given the participant-level parameters, whereas the dynamic model assumes that the session-level parameters have structured temporal correlations induced by practice and state variables (valence and arousal).

To illustrate the advantage of accounting for the full hierarchical structure of the data (sessions nested within participants), in Supplementary Information we provide results based on a third, reduced model that only considers a single level of hierarchy. This reduced model treats the parameters of all sessions as independent draws from a single group-level prior distribution.

**Independent model.** In the independent model, the parameter values across sessions for a given participant were sampled independently around a participant-specific baseline:

$$y_t^s \sim \mathcal{N}\left(\mu^s, \sigma\right), \tag{1}$$

where $y_t^s$ is the phenotype for participant $s$ at time $t$, defined in the unconstrained space (that is, before appropriate transformation was applied to ensure positivity or other constraints; see Supplementary Information for details), and $\sigma$ is the standard deviation (that is, the dispersion of session-level parameters around the participant-level mean). The participant-level mean $\mu^s$ was sampled from a normal distribution at the group level:

$$\mu^s \sim \mathcal{N}\left(\mu^g, \eta_g\right). \tag{2}$$

Notice that the session-level standard deviation, $\sigma$, was defined at the group level. The sampled phenotype value for participant $s$ and session $t$ is therefore the sum of two terms:

$$y_t^s = \mu^s + \sigma_t^s, \tag{3}$$

where $\sigma_t^s$ is the sampled noise term for the session (which can be negative).

The group-level parameters $\sigma, \mu^g, \eta_g^2$ were given weakly informative hyper-priors:

$$\sigma \sim \mathcal{N}_+(0,1) \tag{4}$$

$$\mu^g \sim \mathcal{N}(0,1) \tag{5}$$

$$\eta_g \sim \mathcal{N}_+(0,1) \tag{6}$$

where $\mathcal{N}_+(0,1)$ denotes the half-normal distribution.

**Reduced model.** The reduced model is similar to the independent model, except that it is missing the participant-level priors. The parameter values across all sessions in the dataset were sampled independently around the group baseline:

$$y_t^s \sim \mathcal{N}(\mu^g, \sigma). \tag{7}$$

**Dynamic model.** In the dynamic model, the parameter values across sessions were governed by practice and state effects:

$$y_t^s \sim \mathcal{N}\left(\mu^s + \delta_p^s\left(1 - t^{-b^s}\right) + \delta_v^s v_t^s + \delta_a^s a_t^s, \sigma\right), \tag{8}$$

a sum of the following terms:

- Baseline term: $\mu^s$ is the participant-specific baseline.
- Practice term: $\delta_p^s(1 - t^{-b^s})$ is a plateauing power law. $\delta_p^s \in (-\mu^s, \mu^s)$ is the maximal possible change due to practice, modelled as a fraction of the participant-specific baseline. This fraction was fixed across participants. To promote parameter identifiability, we constrained the learning exponent to be $e^{-2} < b^s < e^1$ (otherwise, the learning curve could flatten and trade off with the baseline term).
- State terms: $\delta_v^s$ and $\delta_a^s$ are the maximal possible changes due to valence and arousal effects, respectively. Similar to the practice term, they were modelled as fixed fractions of the baseline term $\mu^s$. The state predictors $v_t^s$ and $a_t^s$ were normalized within participant, to vary between −1 and 1 across sessions. This reflects the assumption that the phenotype of each participant is affected by the participant-specific state fluctuations around

his or her own baseline, and ensures that the scale of the state terms is governed by $\delta_v^s$ and $\delta_a^s$.
- Noise term: $\sigma^s$ is the participant-specific noise.

The sampled phenotype value for participant $s$ and session $t$ is therefore the sum of five terms:

$$y_t^s = \mu^s + \delta_p^s\left(1 - t^{-b^s}\right) + \delta_v^s v_t^s + \delta_a^s a_t^s + \sigma_t^s. \tag{9}$$

In addition to the priors used in the independent model above, we specified each dynamic parameter $\delta_i$ and their weakly informative priors as follows:

$$\delta_i^s = \delta_i \mu^s \tag{10}$$

$$\delta_i = 2\Phi\left(\delta_i^{\text{prior}}\right) - 1 \tag{11}$$

$$\delta_i^{\text{prior}} \sim \mathcal{N}(0,1) \tag{12}$$

This formulation ensures that $\delta_i \in (-1,1)$, such that each dynamic term can either increase or decrease the phenotype $\mathbf{y}_t^s$.

**Parameter estimation.** Parameters were estimated using a Markov chain Monte Carlo algorithm (we used the default No-U-Turn sampler) implemented in the Stan software package[81]. For each model, we ran 4 chains with 1,000 warmup iterations and 1,000 kept iterations. We used default sampling parameters and initialized each parameter to zero in the unconstrained space. We used the posterior mean for each session and participant as a point estimate of the dynamic computational phenotype.

**Posterior predictive checks.** To check whether our computational models adequately matched participants' behaviour, we simulated data from the fitted models and compared it to behaviour on each task (Fig. 3a and Supplementary Fig. 8). For each task, we generated a figure that captures the key behavioural signatures and compared the empirical data (averaged across participants and sessions) with the predicted data (averaged across participants, sessions and posterior samples).

**Parameter identifiability.** To establish parameter identifiability, we refit the simulated data and checked the degree of match (as measured by correlation) between the recovered and ground-truth parameters. To generate the simulated data, we used the same trial data as presented to the participants, with the following exceptions. The outcomes (rewards/punishments) in the Go/No-go task and the Two-armed bandit task were simulated on the basis of the agent's choices. For these two tasks, we drew the trial-wise outcomes from the same distributions as those presented to the participants, resulting in somewhat different trials than those presented to the participants.

**Parameter stability.** To estimate the stability of each phenotype parameter over time, we followed ref. 24 and calculated non-parametric bootstrapped intraclass correlations, ICC(2,1), with 1,000 samples (bootstrapping over participants)[82–85]. For each phenotype parameter, we used the mean of the posterior distribution as a point estimate and calculated the ICC on the basis of a 90 × 12 data matrix (90 participants with 12 sessions each)[32,34]. However, as noted earlier, some participants had missing sessions and ICC cannot be calculated with missing data. Therefore, for each task, we calculated the ICC only on the basis of participants with complete data. After incorporating the behavioural exclusion criteria (see above), this yielded the following number of participants per task: Go/No-go 53, Change detection 65, Intertemporal choice 67, Lottery ticket 47, Numerosity comparison 69, Two-armed

bandit 66, Random dot motion 73. To verify that this methodological choice did not impact our main results, we repeated the ICC calculation with a different approach. This time, we used data from all participants but using only the maximal number of sessions, $N_{max}$, that existed for all participants. For participants with more sessions, we used the first $N_{max}$ sessions of each participant. The $N_{max}$ values were: Go/No-go 9, Change detection 8, Lottery ticket 8, Two-armed bandit 8, Random dot motion 9, Intertemporal choice 7 and Numerosity comparison 8.

Since we were interested in quantifying the stability of the computational phenotype, it is important to rule out the possibility that methodological considerations, rather than true variability over time, drove the ICC to lower values. For example, even a stable phenotype might seem unstable if not enough trials are used to estimate the computational parameters. To estimate an upper bound for the ICC values given our experimental design, we simulated data from an idealized case in which the distribution of parameter values is the same as in our empirical data, but each agent has a perfectly stable phenotype that remains unchanged throughout the 12 simulated sessions. Specifically, we simulated data from 90 agents, on the basis of the phenotype values of the first session in the empirical data. Then, we refit the independent model to the simulated data to obtain new estimated parameters (which should now be stable over sessions) and calculated the ICC for each parameter. In addition to the ICC of the computational phenotype, we also calculated the ICC values of behavioural summary statistics, namely accuracy and mean reaction time (Supplementary Fig. 5).

**Relative contribution and probability of direction.** To estimate the relative contribution of each dynamical term to the final phenotype, we used the different terms of the dynamic model in the unconstrained space. For simplicity, we used the posterior mean of each term for each participant, yielding a time series of up to 12 timepoints per term. As a measure of amplitude, we calculated the s.d.$_k$ of each term $k$ across sessions. The relative contribution of term $k$ to the final phenotype was defined as: s.d.$_k / \sum_{i=1}^{K}$ s.d.$_i$. Intuitively, dynamical terms with a wider spread contribute more to the variability of the phenotype over time.

Rather than using a binary test such as exclusion of zero from the highest density interval, we chose to quantify the evidence supporting each dynamical effect using the probability of direction (PD). PD is the percentage of posterior probability that has the same sign as the median[42,86].

### Reporting summary

Further information on research design is available in the Nature Portfolio Reporting Summary linked to this article.

## Data availability

The data used in this study are freely available at https://github.com/roeysc/dynamic_computational_phenotyping/data.

## Code availability

The custom code used in this study is freely available at https://github.com/roeysc/dynamic_computational_phenotyping/code.

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

## Acknowledgements

We thank S. Linderman, L. Zhang, L. Fontanesi and P. Kraemer for helpful guidance and discussion; the Stan community (https://discourse.mc-stan.org/), particularly A. Vehtari and S. Pinkney, for addressing our Stan-related questions; J. Mikhael, W. Chen and J. Zou for their contributions. R.S. was supported by the Human Frontier Science Program fellowship LT0046/2022-L. D.R. was supported by the Max Planck Society. R.B. acknowledges support from the Office of Naval Research under Grant No. N00014-21-1-2170. This material is based upon work supported by the National Science Foundation under Grant No. DRL-2024462 and the Zuckerman STEM Leadership Program. The funders had no role in study design, data collection and analysis, decision to publish or preparation of the manuscript.

## Author contributions

R.S., D.R., H.H. and S.J.G. conceived and designed the experiments. D.R. and H.H. performed the experiments. R.S., D.R., R.B. and S.J.G. analysed the data. R.S., D.R., R.B., H.H. and S.J.G. wrote the paper.

## Funding

## Competing interests

The authors declare no competing interests.

## Additional information

**Correspondence and requests for materials** should be addressed to Roey Schurr or Daniel Reznik.

# Reporting Summary

## Statistics

For all statistical analyses, confirm that the following items are present in the figure legend, table legend, main text, or Methods section.

| n/a | Confirmed | |
|---|---|---|
| ☐ | ☒ | The exact sample size (*n*) for each experimental group/condition, given as a discrete number and unit of measurement |
| ☐ | ☒ | A statement on whether measurements were taken from distinct samples or whether the same sample was measured repeatedly |
| ☐ | ☒ | The statistical test(s) used AND whether they are one- or two-sided *Only common tests should be described solely by name; describe more complex techniques in the Methods section.* |
| ☐ | ☒ | A description of all covariates tested |
| ☒ | ☐ | A description of any assumptions or corrections, such as tests of normality and adjustment for multiple comparisons |
| ☐ | ☒ | A full description of the statistical parameters including central tendency (e.g. means) or other basic estimates (e.g. regression coefficient) AND variation (e.g. standard deviation) or associated estimates of uncertainty (e.g. confidence intervals) |
| ☒ | ☐ | For null hypothesis testing, the test statistic (e.g. *F*, *t*, *r*) with confidence intervals, effect sizes, degrees of freedom and *P* value noted *Give P values as exact values whenever suitable.* |
| ☐ | ☒ | For Bayesian analysis, information on the choice of priors and Markov chain Monte Carlo settings |
| ☐ | ☒ | For hierarchical and complex designs, identification of the appropriate level for tests and full reporting of outcomes |
| ☒ | ☐ | Estimates of effect sizes (e.g. Cohen's *d*, Pearson's *r*), indicating how they were calculated |

*Our web collection on statistics for biologists contains articles on many of the points above.*

## Software and code

Policy information about availability of computer code

| Data collection | Amazon Mechanical Turk |
|---|---|
| Data analysis | Python 3.9.0, MATLAB 2019b, R 4.2.3, Stan 2.27, CmdStanPy 0.9.77. <br><br> Code availability <br> The custom code used in this study is freely available at https://github.com/roeysc/dynamic_computational_phenotyping/tree/main/code. |

For manuscripts utilizing custom algorithms or software that are central to the research but not yet described in published literature, software must be made available to editors and reviewers. We strongly encourage code deposition in a community repository (e.g. GitHub). See the Nature Portfolio guidelines for submitting code & software for further information.

## Data

Policy information about availability of data

All manuscripts must include a data availability statement. This statement should provide the following information, where applicable:

- Accession codes, unique identifiers, or web links for publicly available datasets
- A description of any restrictions on data availability
- For clinical datasets or third party data, please ensure that the statement adheres to our policy

> Data availability
> The data used in this study are freely available at https://github.com/roeysc/dynamic_computational_phenotyping/tree/main/data.

## Research involving human participants, their data, or biological material

Policy information about studies with human participants or human data. See also policy information about sex, gender (identity/presentation), and sexual orientation and race, ethnicity and racism.

| | |
|---|---|
| Reporting on sex and gender | Out of the participants that were included in the analyses, 47 identified themselves as males, 41 identified as females, 1 identified as queer. No sex/gender-related hypotheses were tested in the current study as we considered groups effects only. Future studies can potentially explore whether the effects reported in the current study differ between groups. |
| Reporting on race, ethnicity, or other socially relevant groupings | No socially relevant categorization variables were used in the current study. |
| Population characteristics | We recruited adult (>18 years old) Amazon Mechanical Turk participants registered in the USA. The participant pool was filtered to have an approval rating of 90% or above In order to increase the likelihood for continued participation throughout the study. More detailed information on the demographics of Amazon Mechanical Turk participants pool can be found in https://crowdsourcing-class.org/readings/downloads/platform/demographics-of-mturk.pdf . |
| Recruitment | Participants were recruited from Amazon Mechanical Turk through CloudResearch services. Due to the longitudinal nature of our study, participants were rewarded with monetary bonuses after completing half of the study ($5) and the entire study ($10). Therefore, it is possible that our participants were the ones who are particularly interested in increasing their compensation. However, since our rigid selection criterion was approval rating of 90% or above, we do not believe that this monetary-based potential self-selection bias influenced participants' task performance or affected our results. |
| Ethics oversight | Participants read the study description via Amazon Mechanical Turk and indicated whether they agree to participate with a button press. The study was approved by the Institutional Review Board of Harvard University and was performed in accordance with the relevant guidelines and regulations. |

Note that full information on the approval of the study protocol must also be provided in the manuscript.

# Field-specific reporting

Please select the one below that is the best fit for your research. If you are not sure, read the appropriate sections before making your selection.

☐ Life sciences  ☒ Behavioural & social sciences  ☐ Ecological, evolutionary & environmental sciences

For a reference copy of the document with all sections, see nature.com/documents/nr-reporting-summary-flat.pdf

# Behavioural & social sciences study design

All studies must disclose on these points even when the disclosure is negative.

| | |
|---|---|
| Study description | We performed a quantitative study in which human participants performed computer-based tasks. |
| Research sample | Participants were recruited from Amazon Mechanical Turk through CloudResearch services. Our longitudinal design involved multi-day testing each week (during 12 weeks). Therefore we recruited participants online to allow for temporal flexibility. Ninety participants were included in the final analysis (mean age: 39.4 ± 10.8 years; 47 identified as males, 41 identified as females, 1 identified as queer). The male/female ratio in our sample was nearly 0.5; the mean age of USA residents in 2020 was 38.6 (www.statista.com), which is only slightly lower than the mean age in the current sample. |
| Sampling strategy | The sample size (N=90 participants) was determined based on the size of samples often used in the literature. As shown in a literature review by Karvelis et al. 2022 ("Individual Differences in Computational Psychiatry: A Review of Current Challenges"), most behavioral studies that calculated test-retest reliability of the computational phenotype used 30-60 participants. Since retention in longitudinal studies is often a concern, we recruited 141 participants and ended up with 90 participants who completed the study. We stress here that fitting model parameters using a hierarchical Bayesian framework pools information both across participants and across time (in this longitudinal design). Therefore, we expected a sample size of 90 participants to be more than enough for our |

| | |
|---|---|
| | purposes. As for the sampling strategy, participants were sampled randomly. In other words, we did not use any specific strategy to sample participants through the online platform. |
| Data collection | Participants were recruited from Amazon Mechanical Turk through CloudResearch services and performed all the tasks using their personal computers in their own time, given study limitations. There were no experimental groups in the study and all participants followed the same experimental procedure. Researchers were blind to the study hypotheses during data collection. |
| Timing | Data collection started in December 1st 2019 and ended in May 27 2020. |
| Data exclusions | We only included participants who had at least 6 weeks of task data with no more than three missing consecutive weeks for each task. |
| Non-participation | Out of the initial cohort of 141 participants, 90 participants satisfied these criteria. |
| Randomization | Participants were not allocated into experimental groups. |

# Reporting for specific materials, systems and methods

We require information from authors about some types of materials, experimental systems and methods used in many studies. Here, indicate whether each material, system or method listed is relevant to your study. If you are not sure if a list item applies to your research, read the appropriate section before selecting a response.

## Materials & experimental systems

| n/a | Involved in the study |
|---|---|
| ☒ | Antibodies |
| ☒ | Eukaryotic cell lines |
| ☒ | Palaeontology and archaeology |
| ☒ | Animals and other organisms |
| ☒ | Clinical data |
| ☒ | Dual use research of concern |
| ☒ | Plants |

## Methods

| n/a | Involved in the study |
|---|---|
| ☒ | ChIP-seq |
| ☒ | Flow cytometry |
| ☒ | MRI-based neuroimaging |

## Plants

| | |
|---|---|
| Seed stocks | NA |
| Novel plant genotypes | NA |
| Authentication | NA |

