## [Peer Review File · Nature Human Behaviour]

Peer Review Information

Journal: Nature Human Behaviour

Manuscript Title: Dynamic computational phenotyping of human cognition

Corresponding author name(s): Roey Schurr, Daniel Reznik

Editorial Notes:

This manuscript has been previously reviewed at another journal that is not operating a transparent peer review scheme. This document only contains reviewer comments, rebuttal and decision letters for versions considered at Nature Human Behaviour.

Reviewer Comments & Decisions:

Decision Letter, initial version:

16th November 2023

Dear Dr Schurr,

Thank you once again for your revised manuscript, entitled "Dynamic computational phenotyping of human cognition," and for your patience during the re-review process.

Your manuscript has now been evaluated by two returning reviewers who have seen your paper at Nature. All reviewer feedback is included at the end of this letter. Although the reviewers found your manuscript to have improved during revision, one reviewer raises several remaining concerns, including issues of framing and limitations. We would like to see these addressed in text before offering a formal acceptance in principle.

In sum, we invite you to revise your manuscript taking into account all reviewer We are committed to providing a fair and constructive peer-review process. Do not hesitate to contact us if there are specific requests from the reviewers that you believe are technically impossible or unlikely to yield a meaningful outcome.

We hope to receive your revised manuscript within 4 weeks. I would be grateful if you could contact us as soon as possible if you foresee difficulties with meeting this target resubmission date.

- Include a "Response to the editors and reviewers" document detailing, point-by-point, how you addressed each editor and referee comment. If no action was taken to address a point, you must provide

a compelling argument. This response will be used by the editors and reviewers to evaluate your revision.

- Highlight all changes made to your manuscript or provide us with a version that tracks changes.

[REDACTED]

We look forward to seeing the revised manuscript and thank you for the opportunity to review your work. Please do not hesitate to contact me if you have any questions or would like to discuss these revisions further.

Sincerely,

Arunas Radzvilavicius, PhD
Senior Editor, Nature Human Behaviour
Nature Research

REVIEWER COMMENTS:

Reviewer #1:

Remarks to the Author:

I appreciate the efforts of the authors in addressing my concerns and those of the other reviewers. I believe the paper is converging toward a better version, and most of the authors' arguments seem convincing. I have some further remarks that I believe should be considered to enhance the impact of the paper.

1/ It appears to me that the paper is moving towards a methodological focus. One reason for this is the acknowledged fact that no clear cognitive pattern emerges from the dynamical phenotyping; the directionality and magnitude of the tested effects are not coherent across tasks, parameters, and predictors. Another reason is that the paper lacks external validity measures of their approach (by external validity, I mean showing that accounting for phenotypical drifting increases their capacity to capture meaningful external variance in real-life outcomes or psychiatric traits). In this respect, taking the optimistic point of view that ICC upper bounds of the full model are not artifactual, I nonetheless find it very interesting that the authors show that different fitting procedures produce different ICC. I believe that moving some of this material and information into the main text could be useful and will interest many people (in addition to myself!).

2/ On the same line, the authors (correctly and rightfully) put forward the fact that with this paper, they will be releasing a huge and rich dataset, which I am sure will generate a lot of interest. In that respect,

it could be useful to include in the main text a more thorough description of the dataset, including, for example, ranges of descriptive statistics on several behavioral measures.

3/ The introduction is odd because it is still in its "Nature" version (very short, very high level). I believe that it should be drastically reframed to reflect the new venue and provide more background concerning the problem, the proposed solution, together with a deeper overview of the now relatively vast literature on test-retest reliability.

4/ The authors' response to my point concerning the regression is interesting. I think that a similar question may occur to many other scientists. The authors argue that their approach is better and the regression would not work. It would be nice to formally include this argument and results in the paper as a way to strengthen their approach and case.

5/ While we can agree that their model is generative (in a sense) but not mechanistic, the authors should expand on this point to make the distinction explicit and acknowledge as a limitation this fact and the current lack of a mechanistic account as room for future improvement.

6/ Concerning limitations, the lack of external validity should also be mentioned (see the first point).

Reviewer #2:

Remarks to the Author:

I thank the authors for thoroughly addressing my and other reviewers' comments from the first round. I agree it has strengthened the manuscript. I have no further suggestions.

Minor:

What is the last row of Figure S2?

Author Rebuttal to Initial comments

Referee #1 (Remarks to the Author):

I appreciate the efforts of the authors in addressing my concerns and those of the other reviewers. I believe the paper is converging toward a better version, and most of the authors' arguments seem convincing. I have some further remarks that I believe should be considered to enhance the impact of the paper.

1/ It appears to me that the paper is moving towards a methodological focus. One reason for this is the acknowledged fact that no clear cognitive pattern emerges from the dynamical phenotyping; the directionality and magnitude of the tested effects are not coherent across tasks, parameters, and predictors. Another reason is that the paper lacks external validity measures of their approach (by external validity, I mean showing that accounting for phenotypical drifting increases their capacity to capture meaningful external variance in real-life

outcomes or psychiatric traits). In this respect, taking the optimistic point of view that ICC upper bounds of the full model are not artifactual, I nonetheless find it very interesting that the authors show that different fitting procedures produce different ICC. I believe that moving some of this material and information into the main text could be useful and will interest many people (in addition to myself!).

We thank the Reviewer for expressing interest regarding this point. Following the Reviewer's suggestion, we now highlight this result in the main text and in the revised Figure 2, and point the readers to the Supplementary for further details (line 109-120):

“While these ICC values are imperfect, indicating variability of the measured computational phenotype over time, they are relatively high compared with those often reported in the literature [20]. We suspected that these relatively high values can be attributed to fitting our data using hierarchical Bayesian modeling, which adequately captures the hierarchical structure of the data. Indeed, previous work showed that the fitting procedure has notable effects on parameter stability, whereby hierarchical models that pool information across participants promote parameter stability [20, 33, 34, 35]. To test this hypothesis, we repeated the ICC analysis, this time fitting the behavioral data using a reduced hierarchical model. In this reduced model, sessions were not nested within participants; instead, all sessions across all participants were considered independent, such that model parameters were drawn from a single population-level distribution. As expected, this procedure resulted in lower ICC values across all phenotype parameters (Figure 2, red dots). For further details on this analysis see Supplementary Information.”

2/ On the same line, the authors (correctly and rightfully) put forward the fact that with this paper, they will be releasing a huge and rich dataset, which I am sure will generate a lot of interest. In that respect, it could be useful to include in the main text a more thorough description of the dataset, including, for example, ranges of descriptive statistics on several behavioral measures.

We thank the Reviewer for the positive feedback on making our dataset freely available. Following the Reviewer's comment, we now report summary statistics of the main behavioral measures for each task (see below). Since the detailed description of all seven tasks is quite long, we decided to keep it in the Supplementary Information, leaving a shorter version in the Methods section in the main text. In the same vein, we believe that while these summary statistics indeed could be valuable for other researchers, they might obscure the main results of this work. We therefore believe it would be better to report the table as Supplementary Table 1.

Table 1: Behavioral measures of accuracy and reaction times across task conditions (mean \pm standard deviation).

Task	Condition	Accuracy	RT (s)
Go/No-Go	Go2Win	0.85 \pm 0.12	0.54 \pm 0.05
	No-Go2Win	0.62 \pm 0.28	0.58 \pm 0.05
	Go2Avoid	0.64 \pm 0.29	0.57 \pm 0.06
	No-Go2Avoid	0.84 \pm 0.1	0.60 \pm 0.06
Change detection	No target	0.83 \pm 0.08	1.23 \pm 0.69
	One target	0.68 \pm 0.08	1.16 \pm 0.32
	Set size of 8	0.70 \pm 0.06	1.29 \pm 0.81
Random dot motion	5% coherence	0.61 \pm 0.03	0.84 \pm 0.15
	10% coherence	0.73 \pm 0.06	0.81 \pm 0.15
	35% coherence	0.96 \pm 0.05	0.66 \pm 0.13
	50% coherence	0.98 \pm 0.04	0.63 \pm 0.13
Lottery ticket	Small difference	-	2.91 \pm 0.67
	Medium difference	-	2.77 \pm 0.74
	Large difference	-	2.71 \pm 0.72
Intertemporal choice	Short delay	-	2.02 \pm 0.49
	Medium delay	-	2.09 \pm 0.64
	Long delay	-	2.03 \pm 0.58
Two-armed bandit	Risky-Risky	0.75 \pm 0.03	0.44 \pm 0.17
	Safe-Safe	0.89 \pm 0.02	0.41 \pm 0.14
	Risky-Safe/S-R	0.79 \pm 0.04	0.44 \pm 0.16
Numerosity comparison	-	0.85 \pm 0.04	0.62 \pm 0.07

3/ The introduction is odd because it is still in its "Nature" version (very short, very high level). I believe that it should be drastically reframed to reflect the new venue and provide more background concerning the problem, the proposed solution, together with a deeper overview of the now relatively vast literature on test-retest reliability.

We thank the Reviewer for pointing this out. In the revised manuscript we expanded the introduction to include more relevant background and further added another relevant point in the Discussion.

Revised introduction text (lines 20-39):

"For example, research in the field of computational psychiatry demonstrates that computational

modeling can be particularly insightful for teasing apart different behavioral aspects of mental illness. While the link between anxiety and disrupted decision making is well established [14], characterizing the specific behavioral disruption was accomplished in a study that estimated the computational phenotype in patients diagnosed with pathological anxiety and healthy controls. The study showed that anxiety is specifically associated with reduced risk aversion (indicating less risk-taking), but not loss aversion [15]; see also [16]. Another example of the merits of computational phenotyping comes from developmental science. Previous behavioral studies have shown that children tend to explore more than adults [17]. While this could be explained by generally noisier behavior in children, a computational phenotyping study helped to elucidate this phenomenon, indicating that in fact children rely more on directed, but not random exploration, thereby reducing uncertainty about the environment by choosing high-uncertainty options [18].

Despite the widespread use of computational phenotypes, their interpretation hinges critically on their psychometric properties [19], which remain poorly understood [20]. This issue is even more prominent in longitudinal studies that address changes within individuals over time [21, 22]. However, test-retest reliabilities of the computational phenotypes remain largely unknown since computational models are rarely fit within the same subjects over more than one time point. Only a few studies explicitly address the reliability of computational phenotypes (see review in [23]), and rarely in more than two sessions (but see [24]). Such studies have found mixed results, with most phenotype parameters showing poor test-retest reliability, and a few showing moderate to high test-retest reliability. Furthermore, a large-scale study, focusing on the domain of self-control, showed significantly lower reliability for task-based measurements, including the computational phenotype, compared to classic self-reported measurements [25].”

Revised Discussion text (lines 430-436):

“Structural and functional neural states could also be longitudinally recorded concurrently with task performance to increase modeling accuracy in future studies [67]. Neural data can be used to associate the computational phenotype with specific biological phenotypes [68]. Although collecting reliable neurophysiological daily point-estimates is a challenging task, recent advances in 434 individualized brain imaging make this direction practically feasible [69, 70, 71, 72, 73] and particularly exciting in the context of translational neuroscience, potential for clinical application, and precision medicine [74, 75].”

4/ The authors' response to my point concerning the regression is interesting. I think that a similar question may occur to many other scientists. The authors argue that their approach is better and the regression would not work. It would be nice to formally include this argument and results in the paper as a way to strengthen their approach and case.

We thank the Reviewer for this suggestion. As we explain in the revised text, it would be hard to justify the use of a linear regression model in this case, as it would not support strong inference of the parameter values (of the learning curve, for example). Hence, we did not analyze the data

using linear regression. We addressed this point explicitly in Discussion of the revised manuscript (lines 448-457):

“Different statistical approaches could be used to study the effects that influence the dynamics of the computational phenotype. We chose to use a generative model that makes explicit assumptions about the functional form of these dynamical effects in a Bayesian framework. The main advantages of this approach are in providing the full posterior distribution of the model parameters, and in providing a suitably flexible hypothesis space supporting stronger inference. This is in contrast to techniques such as linear regression, which is less suitable in this case, since its hypothesis space is too restrictive. Consider for example the well established power law form of practice effects [40, 41]. While our generative model allows fitting of power law parameters for each participant, a linear regression model would require defining a fixed practice curve across the population, with the regression coefficient as a single free parameter that scales this curve.”

5/ While we can agree that their model is generative (in a sense) but not mechanistic, the authors should expand on this point to make the distinction explicit and acknowledge as a limitation this fact and the current lack of a mechanistic account as room for future improvement.

We thank the Reviewer for this suggestion. We have added the following paragraph to the Discussion (lines 466-473):

“The modeling framework we present here combines task-specific computational models with a general dynamic model of the computational phenotype. Together, they comprise a generative model in the sense that given model parameters, one can produce a time-series of phenotype values (with practice effects and affective effects governed by the model parameters) and simulate behavioral data [77]. This generative model, however, is not mechanistic [78]. While we hint at specific relations between certain state effects (e.g., affective arousal) and phenotype parameters (e.g., risk attitude), understanding the mechanism that underlies these relationships is an open question in cognitive neuroscience.”

6/ Concerning limitations, the lack of external validity should also be mentioned (see the first point).

We thank the Reviewer for this comment. We now mention the fact that we have not examined external validity as one of the limitations of this work (lines 474-479):

“Establishing the reliability of the computational phenotype is crucial if one seeks to study human cognition through computational models of behavior. As illustrated by Karvelis and colleagues [20], characterizing parameter reliability is only the first step in a hierarchy of steps, which was beyond the scope of our current research. Next important step in laying the psychometric foundation of the computational phenotypes would be to establish their external validity and to compare it with the external validity of simpler summary statistics of task

performance [25].

Reviewer #2 (Remarks to the Author):

I thank the authors for thoroughly addressing my and other reviewers' comments from the first round. I agree it has strengthened the manuscript. I have no further suggestions.

Minor:

What is the last row of Figure S2?

We thank the Reviewer for catching this mistake. Please see fixed Figure S2 below:

Decision Letter, first revision:

12th December 2023

Dear Dr. Schurr,

Thank you for your patience as we've prepared the guidelines for final submission of your Nature Human Behaviour manuscript, "Dynamic computational phenotyping of human cognition" (NATHUMBEHAV-23092892A). Please carefully follow the step-by-step instructions provided in the attached file, and add a response in each row of the table to indicate the changes that you have made. Please also address the additional marked-up edits we have proposed within the reporting summary.

Ensuring that each point is addressed will help to ensure that your revised manuscript can be swiftly handed over to our production team.

We would hope to receive your revised paper, with all of the requested files and forms within two-three weeks. Please get in contact with us if you anticipate delays.

Nature Human Behaviour offers a Transparent Peer Review option for new original research manuscripts submitted after December 1st, 2019. As part of this initiative, we encourage our authors to support increased transparency into the peer review process by agreeing to have the reviewer comments, author rebuttal letters, and editorial decision letters published as a Supplementary item. When you submit your final files please clearly state in your cover letter whether or not you would like to participate in this initiative. Please note that failure to state your preference will result in delays in accepting your manuscript for publication.

In recognition of the time and expertise our reviewers provide to Nature Human Behaviour's editorial process, we would like to formally acknowledge their contribution to the external peer review of your manuscript entitled "Dynamic computational phenotyping of human cognition". For those reviewers who give their assent, we will be publishing their names alongside the published article.

Cover suggestions

We welcome submissions of artwork for consideration for our cover. For more information, please see our https://www.nature.com/documents/Nature_covers_author_guide.pdf target="new"> guide for cover artwork.

ORCID

Non-corresponding authors do not have to link their ORCIDs but are encouraged to do so. Please note that it will not be possible to add/modify ORCIDs at proof. Thus, please let your co-authors know that if they wish to have their ORCID added to the paper they must follow the procedure described in the following link prior to acceptance:

Nature Human Behaviour has now transitioned to a unified Rights Collection system which will allow our Author Services team to quickly and easily collect the rights and permissions required to publish your work. Approximately 10 days after your paper is formally accepted, you will receive an email in providing you with a link to complete the grant of rights. If your paper is eligible for Open Access, our Author Services team will also be in touch regarding any additional information that may be required to arrange payment for your article.

Please note that *Nature Human Behaviour* is a Transformative Journal (TJ). Authors may publish their research with us through the traditional subscription access route or make their paper immediately open access through payment of an article-processing charge (APC). Authors will not be required to make a final decision about access to their article until it has been accepted. Find out more about Transformative Journals

[REDACTED]

Best regards,
Alex McKay
Editorial Assistant
Nature Human Behaviour

On behalf of

Arunas Radzvilavicius, PhD
Senior Editor, Nature Human Behaviour
Nature Research

Final Decision Letter:

Dear Dr Schurr,

We are pleased to inform you that your Article "Dynamic computational phenotyping of human cognition", has now been accepted for publication in Nature Human Behaviour.

Please note that *Nature Human Behaviour* is a Transformative Journal (TJ). Authors may publish their research with us through the traditional subscription access route or make their paper immediately open access through payment of an article-processing charge (APC). Authors will not be required to make a final decision about access to their article until it has been accepted. Find out more about Transformative Journals

Once your manuscript is typeset and you have completed the appropriate grant of rights, you will receive a link to your electronic proof via email with a request to make any corrections within 48 hours. If, when you receive your proof, you cannot meet this deadline, please inform us at

rjsproduction@springernature.com immediately. Once your paper has been scheduled for online publication, the Nature press office will be in touch to confirm the details.

With best regards,

Arunas Radzvilavicius, PhD
Senior Editor, Nature Human Behaviour
Nature Research